# Advancing Self-supervised Monocular Depth Learning with Sparse LiDAR

**Ziyue Feng**[1]   **Longlong Jing**[2]   **Peng Yin**[3]   **Yingli Tian**[2]   **Bing Li**[1]

[1]Clemson University   [2]The City University of New York   [3]Carnegie Mellon University

**Abstract:** Self-supervised monocular depth prediction provides a cost-effective solution to obtain the 3D location of each pixel. However, the existing approaches usually lead to unsatisfactory accuracy, which is critical for autonomous robots. In this paper, we propose FusionDepth, a novel two-stage network to advance the self-supervised monocular dense depth learning by leveraging low-cost sparse (e.g. 4-beam) LiDAR. Unlike the existing methods that use sparse LiDAR mainly in a manner of time-consuming iterative post-processing, our model fuses monocular image features and sparse LiDAR features to predict initial depth maps. Then, an efficient feed-forward refine network is further designed to correct the errors in these initial depth maps in pseudo-3D space with real-time performance. Extensive experiments show that our proposed model significantly outperforms all the state-of-the-art self-supervised methods, as well as the sparse-LiDAR-based methods on both self-supervised monocular depth prediction and completion tasks. With the accurate dense depth prediction, our model outperforms the state-of-the-art sparse-LiDAR-based method (Pseudo-LiDAR++ [1]) by more than 68% for the downstream task monocular 3D object detection on the KITTI Leaderboard. Code is available at https://github.com/AutoAILab/FusionDepth

**Keywords:** Self-supervised, Monocular, Depth Prediction, Sparse LiDAR

## 1   Introduction

Obtaining the 3D location of objects is an essential task for autonomous robots. However, accurate dense depth perception with LiDAR is normally expensive, thus limit it for mass production. The depth prediction from monocular images [2, 3] is cost-effective and attracting more and more attention from both research and industry communities.

Many methods have been proposed and remarkable progress has been achieved in recent years [4, 5, 6, 7]. Unlike other computer vision tasks, it is impractical to obtain large-scale dense depth labels. Therefore, self-supervised monocular depth prediction has been a promising solution. Typically networks are trained to predict both the depth and ego-motion of the camera, while the re-projection photo-metric loss is calculated as an intermediary constraint to optimize the networks. However, these methods usually suffer from multiple challenges due to the loss function design. The most significant one is that the re-projection constraint assumes the scene is static and without occlusions between the neighboring frames. In fact, most of the vital objects (e.g., vehicles, pedestrians, and cyclists in the driving scenario) are dynamic, and the occlusions are almost inevitable.

To handle these challenges, a potential direction is to perform the monocular depth prediction with other low-cost sensors like sparse (e.g. 4-beam) LiDAR. Compared to the 64-beam LiDAR, the cost of 4-beam LiDAR is at least two orders lower while providing very sparse yet accurate depth. It is too sparse to be directly used for high-level perception tasks but potentially can be used with images to guide the network for better dense depth prediction Pseudo-LiDAR++ [1] employed the sparse LiDAR in the post-processing using a graph-based depth correction (GDC) module to improve the performance of stereo 3D detection. However, this approach has two significant limitations: 1) the quality of its predicted dense depth is non-optimal since the sparse LiDAR data is not utilized in the depth prediction network; 2) the GDC post-processing

*Corresponding author: Bing Li <bli4@clemson.edu>

5th Conference on Robot Learning (CoRL 2021), London, UK.

did not utilize the visual context information, and is too slow (1 to 2 FPS) for real-time applications like autonomous robots. Pursing an accurate and real-time self-supervised monocular depth prediction, we propose a two-stage network for depth prediction by fully utilizing the sparse LiDAR points and monocular images in both the feature and the prediction levels. Our framework can learn the complementary information from the two distinct types of features for the dense depth prediction tasks. To overcome the sparsity issue of the sparse LiDAR, we transform the sparse LiDAR points into pseudo dense representations, which are more suitable for networks to extract features, and then fuse the features with the image features to predict the initial depth. To further improve the quality of initial depth, we train a RefineNet to efficiently correct the high-order errors in the 3D space to obtain high-quality dense depth maps.

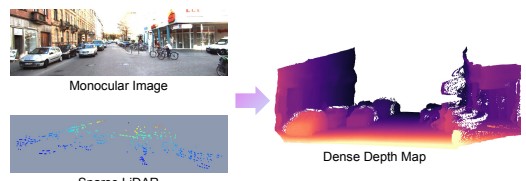

Figure 1: The sparse LiDAR (e.g. 4-beam) provides sparse yet accurate points which cannot be directly used for high-level downstream tasks such as detection due to the sparsity. The monocular depth prediction can be significantly improved by effectively fusing the features of the sparse LiDAR via self-supervised learning. With our high-quality predicted depth, the performance of the downstream perception tasks such as monocular 3D object detection can be remarkably improved.

The ultimate goal of depth prediction is to provide 3D information for downstream tasks such as monocular 3D detection and re-construction. However, the relation between the performance of depth prediction and the high-level downstream tasks has not yet been explored. To thoroughly evaluate our proposed method, we report the performance for both low-level tasks, including self-supervised dense depth prediction and completion, and a downstream high-level perception task: monocular 3D object detection. On all these tasks, our proposed model **FusionDepth** significantly outperforms the state-of-the-art methods that rely on or do not rely on sparse LiDAR. To summarize, our key contributions are as follows:

- We propose a novel two-stage self-supervised network to predict and refine dense depth maps by fusing the features of 2D monocular images and 3D sparse LiDAR points. Our experiments demonstrates that our model achieves state-of-the-art performance in the depth prediction and completion tasks on the KITTI dataset.

- To overcome the sparsity issue of the sparse LiDAR, we propose to transform the sparse points into a novel pseudo dense representation (PDR) which can be more effectively fused with monocular image features.

- With the improved predicted depth maps, the performance of the downstream task monocular 3D object detection is significantly improved. Our model outperforms the state-of-the-art sparse-LiDAR-based 3D detection model (Pseudo-LiDAR++ [1]) by more than 68% on the KITTI dataset.

## 2 Related Work

**Self-supervised Monocular Depth Prediction**: Early work for depth prediction is usually supervised methods [4, 3, 8, 9, 10, 11, 5, 6, 7, 12]. Pixel-level labeled dense depth is rarely available, in recent years, self-supervised methods [13, 14, 15, 16] became more and more popular. They have achieved great success but still suffer from dynamic objects and scale ambiguity. In contrast, we fuse the feature from sparse LiDAR points to help our method predict a more accurate depth for each pixel.

**Depth Prediction with Sparse LiDAR**: Recently, many researchers proposed to use few-beam LiDAR for better depth prediction [17, 18, 19, 20, 1]. The Pseudo LiDAR++ [1] achieved excellent performance by a GDC post-processing module to optimize the predicted depth with 4-beam LiDAR data. However, the potential of the sparse LiDAR is not fully discovered since the sparse LiDAR points are unused in the initial depth prediction stage. To effectively utilize the sparse LiDAR and monocular images, we fuse them in both feature and prediction level for more accurate depth predictions.

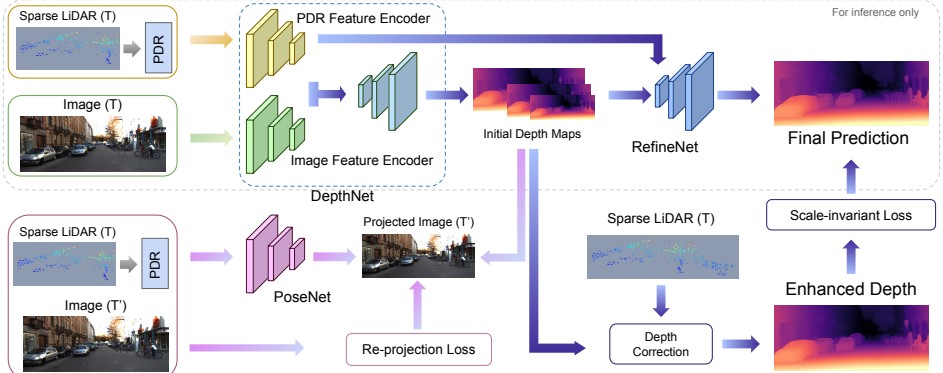

Figure 2: **Overview of our framework:** Our proposed two-stage self-supervised FusionDepth model takes a monocular image and the corresponding sparse LiDAR points as input, and predicts a dense depth map for each monocular image. At inference time, only the modules inside the gray dashed rectangle are needed.

**Depth Completion**: The depth completion is a task to generate per-pixel dense depth maps from the relative sparse depths. Most depth completion models [21, 22, 23, 24, 25, 26] are trained with labeled data which require intensive human labors. To utilize the massive unlabeled data, self-supervised depth completion methods were developed in recent years [27, 28, 29, 30, 31] to generate depth maps from 64-beams dense LiDAR points. Our proposed method is designed to predict depth maps from 4-beams sparse LiDAR points, however, with the generalizability, our method is also applicable for the depth completion task.

**Monocular 3D Object Detection:** Monocular 3D object detection is to directly predict the 3D coordinates of objects from monocular images. There are mainly two types of methods: RGB image-based and pseudo-LiDAR based. Former employ detection networks like CenterNet [32] to predict the bounding boxes [33, 34, 35, 36] directly from images. The latter perform the detection over the pseudo-LiDAR representation, which is lifted from the dense depth prediction [37, 38, 39, 40]. Benefited from the 2D-to-3D mapping, the pseudo-LiDAR-based methods achieved much better performance [39]. Our experiments demonstrate that the performance of the monocular 3D object detection [1, 39] can be significantly improved using our depth prediction.

## 3  The Proposed Method

The overview of our proposed self-supervised framework is shown in Fig. 2. Our framework predicts a dense depth map for each monocular image by taking two types of data as input: monocular image and its corresponding sparse LiDAR points. Our framework consists of two steps: initial depth prediction based on the fused multi-scale features from both the monocular image and sparse LiDAR points, and depth refinement to correct the high-order errors in the initial depth maps. The details of each component are described in the following sections.

### 3.1  Initial Depth Generation

For each monocular image $I^{H \times W \times 3}$, the corresponding sparse LiDAR points are $P^{N \times 3}$ captured by few-beam LiDAR (e.g. 4-beam) where $N$ is the number of the points. Each point $p_i$ consists of three values $X, Y, Z$, representing the location in the 3D space. The network predicts an initial depth map for each image based on the fusion of features from image $I$ and the corresponding sparse LiDAR points $P$.

The main challenge here is to effectively fuse the features from a 2D image and the features from a set of unordered LiDAR points. When projecting all the sparse LiDAR points into the image plane, only $1.4\%$ of the pixels have corresponding depth values. We observe that simply concatenating the image data and the projected sparse depth map can only negligibly improve the performance due to the sparsity of the representation. To resolve the sparsity issue, we transform the sparse LiDAR

points into pseudo dense representations (PDRs), which can be effectively encoded and fused with the monocular image features by our model.

**Pseudo Dense Representations Generation:** For each image $I^{H \times W \times 3}$, all its corresponding sparse LiDAR points $P^{N \times 3}$ are transformed into two-channel PDRs with a size of $H \times W \times 2$, including a depth channel to present the absolute depth values of each pixel, and a confidence channel presents the reliability of the corresponding depth. Based on the assumption that the depth values should be similar for most neighboring pixels, we generate the depth channel by first projecting each sparse LiDAR point $p_i$ into the image plane at the position of $(u_i, v_i)$ and then dilate it into a circular area with a radius of $R$. The depth channel ($D$) is generated by setting the depth value of pixel $(x, y)$ within this circular area as $Z_i$:

$$r(x, y) = \sqrt{(u_i - x)^2 + (v_i - y)^2} \tag{1}$$

$$D(x, y) = \begin{cases} Z_i, & \text{if } r(x, y) < R \\ 0, & \text{otherwise} \end{cases}. \tag{2}$$

Although the depth channel transfers the sparse LiDAR points into dense representation, it inevitably introduces noises. To mitigate the impact of these noises, we further generate the confidence channel to indicate the reliability of depth for each pixel. The confidence of each pixel is inversely proportional to the distance to its circular center:

$$C(x, y) = \begin{cases} \frac{1}{r}(x, y), & \text{if } r(x, y) < R \\ 0, & \text{otherwise} \end{cases}, \tag{3}$$

If more than one sparse data point is close to an PDR pixel, the confidence and depth scores generated from these multiple data points will be averaged. These two channels jointly provide alternative dense representations that can be encoded by the convolutional neural networks more effectively.

**DepthNet for Initial Depth Map Prediction Based on Fused Features.** After transforming the sparse LiDAR points into the two-channel PDRs, the features of PDRs and monocular images can be fused together for the initial depth prediction. To enable our network to thoroughly learn the complementary information from two distinct features, we adopt the intermediate fusion to combine the multi-scale deep features layer by layer. As shown in Fig. 2, there are two feature encoders: PDR feature encoder for extracting features that explicitly encodes depth information from the sparse LiDAR pseudo dense representation, and monocular image feature encoder to extract feature which implicitly encodes semantic information from images. The decoder network takes the two types of multi-scale features and concatenates them together to predict the initial depth map at multiple scales. With the effective feature fusion, our DepthNet can predict more accurate dense depths.

**PoseNet for Ego-Motion Prediction.** The PoseNet is essential for self-supervised monocular depth prediction since its predicted ego-motion makes the cross-frame projection possible, which is used as geometry constraints to train the network. It is directly related to the quality of the pixel correspondence across frames. To predict accurate ego-motions, as shown in Fig. 2, the PoseNet is also designed to take the complementary monocular images and the PDRs as the input. The ego-motion is formulated as 6 degrees of freedom, consisting the camera rotation and translation.

## 3.2 Depth Refinement

Our network produces relatively accurate initial depth predictions by fusing the features of monocular images and sparse LiDAR points. However, the network still makes various errors, such as inconsistent depth prediction for different parts of the same object and the systematic depth bias.

To further resolve the inconsistent depth prediction and improve the correction computing efficiency, we propose a RefineNet to correct the initial depth errors in the pseudo 3D space [39]. Our RefineNet is a multi-scale fully convolutional network, takes the PDR feature, image feature, and initial depth as input, outputs the refined final depth prediction. To leverage the 3D information predicted from DepthNet, each initial depth map is first transformed into a 3-channel x-y-z map representation. For a pixel at location $(u, v)$ with depth $d$ at the initial depth map, the transformation is based on the camera intrinsics $f_x, f_y, C_x, C_y$:

$$\begin{cases} x = (u - C_x) \times d/f_x, \\ y = (v - C_y) \times d/f_y, \\ z = d, \end{cases} \tag{4}$$

Our RefineNet is trained by distilling knowledge from a offline depth correction module in the way of self-training. As shown in Fig. 2, for each initial depth map, a depth correction module is applied to produce a more accurate depth named 'Enhanced Depth' with the guidance of sparse LiDAR points. By training with pairs of the pseudo 3D initial depth, PDR and image features, and the enhanced depth, the RefineNet can improve the initial depth quality by correcting its errors. Compared to the depth correction module GDC [1] we used as teacher module, our network is very computationally efficient for real-time systems. Moreover, the conventional depth correction module can still be further applied as post-processing after our RefineNet to improve the accuracy by sacrificing the real-time performance.

### 3.3 Loss Functions

Our model is jointly trained with two loss functions: the re-projection loss $L_{re}$ to utilize the inter-frame geometric constraints and the scale-invariant loss $L_{si}$ to distill the knowledge from the enhanced depth maps generated by the offline depth correction model.

The re-projection loss $L_{re}$ is a linear combination of two parts: the photo-metric loss $L_p$ with a filtering mask $\mu$, and the smoothness loss $L_{smooth}$. The photo-metric loss $L_p$ is to evaluate the pixel-level similarity between the re-projected fake image $I_{t \to t'}$ with the real image $I'_t$ at adjacent time frames, based on the photo-metric reconstruction error $pe$ which consists of the SSIM and $L1$ distance to penalize the errors of the re-projected image. We choose to use the frames $I_{t+1}$ and $I_{t-1}$ as $I_{t'}$. The $proj()$ projects current frame dense depth $D_t$ to frame $t'$ with camera intrinsic matrix $K$ and the camera ego-motion estimated by the PoseNet, $\langle \rangle$ is the sampling operator.

$$L_{re} = \mu L_p + L_{smooth}, \tag{5}$$

$$L_p = \sum_{t'} pe(I_t, I_{t \to t'}), \tag{6}$$

$$I_{t \to t'} = I_t \left\langle proj(D_t, T_{t \to t'}, K) \right\rangle, \tag{7}$$

$$pe(I_a, I_b) = \frac{\gamma}{2}(1 - \text{SSIM}(I_a, I_b)) + (1 - \gamma)\|I_a - I_b\|_1 \tag{8}$$

To eliminate the shrinking of the depth map, we further adopt the edge-aware metric from [41] into our smoothness loss function $L_{smooth}$, which is formulated as:

$$L_{smooth} = |\partial_x d_t^*| e^{-|\partial_x I_t|} + |\partial_y d_t^*| e^{-|\partial_y I_t|}, \tag{9}$$

while $d_t^* = d_t/\overline{d_t}$ is the mean-normalized inverse depth. This normalization makes the smoothness loss invariant to output scale.

For the photo-metric loss $L_p$, we follow [15] to apply a filtering mask $\mu$ to filter out the occlusion and stationary pixels, and then interpolate the depth predictions at each scales to the input resolution before computing our re-projection loss to eliminate the 'holes' at the low-texture area. The filtering mask $\mu$ is formulated as:

$$\mu = \left[ \min_{t'} pe(I_t, I_{t' \to t}) < \min_{t'} pe(I_t, I_{t'}) \right]. \tag{10}$$

To distill the knowledge from the offline correction model, the scale-invariant loss is employed for optimization, which is formulated as:

$$L_{si} = \lambda * \sqrt{\eta Si}, \tag{11}$$

$$Si = \frac{1}{n^2} \sum_{i,j} \left( (\log y_i - \log y_j) - (\log y_i^* - \log y_j^*) \right)^2, \tag{12}$$

where $y$ and $y^*$ indicate the predicted depth and enhanced depth respectively, $n$ is the number of pixels. The $Si$ loss penalizes the relative depth-differences between each pixel pairs.

Our framework is trained with a linear combination of the re-projection loss $L_{re}$ and the scale-invariant loss $L_{si}$:

$$L_{total} = \alpha L_{re} + \beta L_{si}, \tag{13}$$

while $\alpha$ and $\beta$ are the weights for re-projection loss $L_{re}$ and scale-invariant loss $L_{si}$ respectively. The implementation details can be found in the supplementary materials.

Table 1:

| Method | Train | The lower the better | | | | The higher the better | | |
|---|---|---|---|---|---|---|---|---|
| | | Abs Rel | Sq Rel | RMSE | RMSE log | $\delta_1$ | $\delta_2$ | $\delta_3$ |
| LEGO [42] | M | 0.162 | 1.352 | 6.276 | 0.252 | 0.783 | 0.921 | 0.969 |
| PackNet-SfM [16] | M | 0.111 | 0.785 | 4.601 | 0.189 | 0.878 | 0.960 | 0.982 |
| MonoDepth [14] | S | 0.133 | 1.142 | 5.533 | 0.230 | 0.830 | 0.936 | 0.970 |
| MonoDepth2 [15] | M+S | 0.106 | 0.818 | 4.750 | 0.196 | 0.874 | 0.957 | 0.979 |
| Dorn [43] | M+Sup | 0.099 | 0.593 | 3.714 | 0.161 | 0.897 | 0.966 | 0.986 |
| BTS [44] | M+Sup | 0.091 | 0.555 | 4.033 | 0.174 | 0.904 | 0.967 | 0.984 |
| Guizilini *et al.* [45]* | M+L | 0.082 | 0.424 | 3.73 | 0.131 | 0.917 | - | - |
| **FusionDepth (Initial Depth)** | M+L | 0.078 | 0.515 | 3.67 | 0.154 | 0.935 | 0.973 | 0.986 |
| **FusionDepth (Refined Depth)** | M+L | **0.074** | **0.423** | **3.61** | 0.150 | **0.936** | **0.973** | **0.986** |
| Struct2Depth [46] | M$^\dagger$ | 0.109 | 0.825 | 4.750 | 0.187 | 0.874 | 0.958 | 0.983 |
| GLNet [47] | M$^\dagger$ | 0.099 | 0.796 | 4.743 | 0.186 | 0.884 | 0.955 | 0.979 |
| MonoPL++ [1] | M+L$^\dagger$ | 0.098 | 0.714 | 4.30 | 0.176 | 0.899 | 0.967 | 0.984 |
| **FusionDepth (Initial Depth + GDC)** | M+L$^\dagger$ | 0.067 | 0.423 | 3.42 | 0.144 | 0.941 | 0.977 | 0.988 |
| **FusionDepth (Refined Depth + GDC)** | M+L$^\dagger$ | **0.063** | **0.364** | **3.291** | **0.139** | **0.945** | **0.978** | **0.988** |

Table 1: **Depth prediction on KITTI original dataset:** Methods are ranked by absolute relative error. The best results are in bold. All methods are using a resolution of 640x192 pixels. Due to the exceptional time-consuming (around 1-2 FPS), we rank methods with and without iterative refinement separately. $M$, $S$, and $L$ respectively indicates Monocular, Stereo, and Sparse LiDAR data, with $Sup$ and $\dagger$ respectively indicating supervised training and iterative correction in testing phase. * Only use LiDAR data in training phase, but tested on the KITTI improved dataset, which usually has a much lower error value.

| Method | Samples | The lower the better | | The higher the better | | |
|---|---|---|---|---|---|---|
| | | Abs Rel | RMSE | $\delta_1$ | $\delta_2$ | $\delta_3$ |
| full-MAE [17] | ~650 | 0.179 | 7.14 | 70.9 | 88.8 | 95.6 |
| Liao *et al.* [18] | 225 | 0.113 | 4.50 | 87.4 | 96.0 | 98.4 |
| Sparse2Dense [19] | 100 | 0.095 | 4.303 | 90.0 | 96.3 | 98.3 |
| **FusionDepth** | 100 | 0.074 | **4.11** | **93.0** | **97.0** | 98.3 |
| **FusionDepth + GDC** | 100 | **0.073** | **4.11** | **93.0** | **97.0** | 98.3 |
| Sparse2Dense [19] | 200 | 0.083 | 3.851 | 91.9 | 97.0 | 98.6 |
| **FusionDepth** | 200 | 0.069 | **3.92** | **93.7** | 97.0 | 98.3 |
| **FusionDepth + GDC** | 200 | **0.066** | **3.92** | **93.7** | **97.1** | **98.4** |

Table 2: **Depth prediction with random-sampled LiDAR points:** Comparison of performances on the KITTI dataset [48] with methods that also rely on sparse LiDAR points. The input point clouds are randomly sampled from 64-beam LiDAR points. Our FusionDepth outperforms all other methods with a large gap even without refinement.

| Method | Supervised Depth | KITTI Testing ($AP\|_{40}$) | | |
|---|---|---|---|---|
| | | Easy | Mod. | Hard |
| Pseudo LiDAR++* [1] | ✓ | 68.5 | 54.7 | 51.5 |
| Decoupled-3D [49] | ✓ | 11.08 | 7.02 | 5.63 |
| MonoPSR [50] | - | 10.76 | 7.25 | 5.85 |
| MonoPL [38] | ✓ | 10.76 | 7.50 | 6.10 |
| SS3D [51] | - | 10.78 | 7.68 | 6.51 |
| MonoDIS [52] | - | 10.37 | 7.94 | 6.40 |
| M3D-RPN [53] | - | 14.76 | 9.71 | 7.42 |
| AM3D [54] | ✓ | 16.50 | 10.74 | 9.52 |
| PatchNet [39] | ✓ | 15.68 | 11.12 | 10.17 |
| MonoPL++ [1] | ✗ | 14.93 | 10.85 | 9.50 |
| **FusionDepth(Ours)** | ✗ | **25.21** | **18.99** | **16.53** |
| | | +68.9% | +75.0% | +74.0% |

Table 3: **3D detection performance evaluation** for the **Car** category on the KITTI dataset *testing* set. $AP_{3d}$@0.7.
*The depth module of Pseudo LiDAR++ [1] is stereo, which can greatly improve the detection result. For fair comparison, we replace the it with Monodepth2 [15], and refer as MonoPL++ [1]

## 4 Experiments

### 4.1 Depth Prediction

**Dataset.** Following the state-of-the-art methods [15, 16, 13], we evaluate our FusionDepth performance of dense depth prediction on the Eigen split [55] of the KITTI original dataset. We did not evaluate on the KITTI testing benchmark which is for vision-only methods. The 4-beams data are sampled from original 64-beams LiDAR data same as Pseudo LiDAR++ [1].

**Results.** To extensively evaluate the performance, we compare with four types of methods including: (1) self-supervised monocular-based methods (**M**) [56, 42, 41, 57, 58, 15, 16, 13], (2) self-supervised stereo-based methods (**S**) [59, 60, 14, 61, 62], (3) supervised methods (**Sup**) [43, 44], and (4) methods that use LiDAR signal as guidance (**L**) [1, 45]. The performance comparison with the state-of-the-art methods for these groups is shown in Table 1. Note that the initial depth module in Pseudo LiDAR++ [1] is supervised and stereo. For fair comparison, we replace it with the state-of-the-art unsupervised monocular depth module Monodepth2 [15]. We will refer this model as MonoPL++.

Due to the limitations of the re-project photo-metric loss, the self-supervised monocular (M) [56, 42, 41, 57, 58, 15, 16, 13] and stereo-based methods (S) [59, 60, 14, 61, 62] usually have Abs Rel over 0.1. With the advantage of using the sparse LiDAR, the performance of our initial depth maps

| Method | FPS | mAP Mod. | Car | | | Pedestrian | | | Cyclist | | | Depth Abs.Rel |
|---|---|---|---|---|---|---|---|---|---|---|---|---|
| | | | Easy | Mod. | Hard | Easy | Mod. | Hard | Easy | Mod. | Hard | |
| Monodepth2 [15] | | 6.45 | 17.35 | 12.86 | 11.45 | 5.24 | 4.94 | 4.54 | 2.29 | 1.55 | 1.55 | 10.27 |
| **FusionDepth(Initial Depth)** | **>100** | 8.18 | 22.20 | 15.37 | 14.55 | **6.97** | 6.49 | 5.71 | 4.20 | 2.69 | 2.59 | 7.51 |
| **FusionDepth(Refined Depth)** | | **9.02** | **22.29** | **15.42** | **14.76** | 6.68 | **6.50** | **6.00** | **7.27** | **5.16** | **5.14** | **7.25** |
| Delta (%) | | **+40%** | +28% | +20% | +29% | +33% | +32% | +32% | +217% | +233% | +206% | 29% |
| MonoPL++ (GDC) [1] | | 10.56 | 33.75 | 22.38 | 20.45 | 6.41 | 5.30 | 5.14 | 5.89 | 4.00 | 4.07 | 8.41 |
| **FusionDepth(Initial Depth+GDC)** | 1-2 | 16.94 | 41.53 | 29.49 | 24.29 | 14.81 | 11.97 | 10.53 | 15.24 | 9.35 | 8.91 | 6.39 |
| **FusionDepth(Refined Depth+GDC)** | | **20.93** | **44.55** | **33.59** | **28.87** | **18.28** | **14.46** | **12.32** | **23.28** | **14.75** | **13.38** | **6.17** |
| Delta (%) | | **+98%** | +32% | +50% | +41% | +185% | +172% | +140% | +295% | +268% | +229% | 26% |

Table 4: Monocular 3D object detection result with PatchNet [39] on KITTI dataset, $AP@0.7$ for cars, $AP@0.5$ for pedestrians and cyclists. Our FusionDepth can greatly improve the performance both with or without GDC.

is already better than all these methods and even outperforms the supervised methods (Sup) [43, 44]. With our RefineNet, our performance is further improved and outperforms all the sparse-LiDAR-based methods [1, 45]. Using the GDC for post-processing, our final results significantly outperform all other methods, including the above mentioned most recent work MonoPL++ [1] which has access to the same amount of the sparse LiDAR points and same post-processing. The results indicate an effective fusion of sparse LiDAR points and monocular images achieves more accurate dense depth predictions. More quantitative and qualitative comparison and error analysis can be found in the supplementary materials.

To extensively compare with the state-of-the-art methods under the same settings, as shown in Table 2, we compare with methods that were originally designed to use sparse LiDAR for depth prediction, including full-MAE [17], Parse-a-Line [18], Sparse-to-Dense [19], and MonoPL++ [1]. For each group of experiments, the same amount of sparse LiDAR points is used for a fair comparison. Under the same settings, our proposed model consistently significantly outperforms all the state-of-the-art methods. These results further confirm the effectiveness of our proposed method.

## 4.2 Monocular 3D Object Detection

The ultimate goal of the monocular depth prediction is to provide 3D representation for downstream tasks. To demonstrate the impact of our improvement on the depth metrics to downstream tasks, we evaluate the performance of the monocular 3D object detection task with our predicted depth maps as input on the KITTI detection dataset. Following the state-of-the-art methods [39], we report 3D Average Precision (AP) as the evaluation metrics.

Table 4 shows the performance comparison for the monocular 3D object detection task. The same detection model is employed for all the experiments while the only difference is the input depth. The monodepth2 [15] is used as baseline model. Our model use the same CNN backbone (ResNet-18 [63]) as Monodepth2 [15]. By effectively using the sparse LiDAR, the detection performance can be improved by more than 40% compared to the baseline which only uses monocular images. Compared to the recently proposed MonoPL++ [1], our model significantly outperforms it by more than 98.2% in terms of the mAP over all the three categories. Without the time-consuming iterative refinement module, our model is 50 times faster than the MonoPL++ [1].

Furthermore, our RefineNet obtains a significantly increased performance on the detection metrics than the depth metrics. The improvement of applying RefineNet on the depth score is only around 3.4% in terms of the relative error, while the improvement is more than 23% on the detection score. We observe that our improvement in the dense depth prediction can yield a even more significant improvement in the downstream task. This indicates the importance of using the downstream tasks to evaluate the quality of the learned dense depth.

Table 3 further shows the comparison of our method with other state-of-the-art for monocular 3D detection tasks on the car category. As an unsupervised learning method, our method significantly outperforms all the state-of-the-art methods by a large margin, including the sparse-LiDAR-based MonoPL++ [1]. Note that the performance of our model can be further improved by extending it to the supervised setting. More qualitative analysis can be found in the supplementary materials.

| Pseudo Dense Representation | Camera-LiDAR Fusion in Initial Prediction | | | Refinement | | The Lower the Better | | | |
|---|---|---|---|---|---|---|---|---|---|
| | Input Level | Output Level | Feature Level | RefineNet | GDC | Abs Rel | Sq Rel | RMSE | RMSE$_{log}$ |
| Evaluating Pseudo Dense Representation (PDR) | | | | | | | | | |
| | | | | | | 0.115 | 0.882 | 4.701 | 0.190 |
| | ✓ | | | | | 0.108 | 0.814 | 4.588 | 0.184 |
| ✓ | ✓ | | | | | **0.101** | **0.726** | **4.364** | **0.178** |
| Evaluating Camera-LiDAR Fusion in Initial Prediction | | | | | | | | | |
| ✓ | ✓ | | | | | 0.101 | 0.726 | 4.364 | 0.178 |
| ✓ | | ✓ | | | | 0.115 | 0.907 | 4.847 | 0.192 |
| ✓ | ✓ | ✓ | | | | 0.102 | 0.734 | 4.369 | 0.177 |
| ✓ | | | ✓ | | | **0.078** | **0.515** | **3.678** | **0.154** |
| Evaluating Refinement | | | | | | | | | |
| ✓ | | | ✓ | | | 0.078 | 0.515 | 3.678 | 0.154 |
| ✓ | | | ✓ | ✓ | | 0.074 | 0.433 | 3.610 | 0.150 |
| ✓ | | | ✓ | | ✓ | 0.067 | 0.425 | 3.420 | 0.144 |
| ✓ | | | ✓ | ✓ | ✓ | **0.063** | **0.346** | **3.291** | **0.139** |

Table 5: **Ablation Study:** Results on the KITTI depth prediction dataset Eigen [55] split. We evaluate the effectiveness of PDR, Camera-LiDAR fusion, and RefineNet.

## 4.3 Depth Completion

We evaluate our FusionDepth on the KITTI depth completion task. Most (over 95%) of the methods in the KITTI depth completion benchmark is under supervised training. Following other state-of-the-art self-supervised methods [27, 28, 29, 30, 31], we test our model on the KITTI validation set, as shown in Table 6. By effectively utilizing the LiDAR features, our model outperforms all other self-supervised methods by a large gap with all the metrics demonstrating the generalizability of our proposed method.

## 4.4 Ablation Study

To evaluate the effectiveness of each component of our FusionDepth, we perform three groups of experiments to evaluate the impact of pseudo dense representation, camera-LiDAR fusion, and the refinement on the dense depth prediction task. The results of the ablation studies are shown in Table 5. The impact of the LiDAR sparsity and the error analysis by semantic categories can be found in the supplementary materials.

| Method | RMSE | iRMSE | iMAE |
|---|---|---|---|
| KISS-GP [31] | 1593.37 | 27.98 | 2.36 |
| Sparse to dense [28] | 1342.33 | 4.28 | 1.64 |
| DDP [30] | 1310.03 | - | - |
| VOICED [29] | 1230.85 | 3.84 | 1.29 |
| SelfDeco [27] | 1212.89 | 3.54 | 1.29 |
| **FusionDepth(Ours)** | **1193.92** | **3.385** | **1.28** |

Table 6: **Self-supervised depth completion:** We evaluate our FusionDepth on the KITTI depth completion task validation set, comparing it to the state-of-the-art self-supervised methods. All metrics are the lower, the better. The best results are in bold.

**Pseudo Dense Representation.** When the sparse LiDAR are directly used as input, the performance for depth prediction is only slightly improved by around 6%. However, by directly using our proposed pseudo dense presentation as input, the improvement is doubled (12%), confirming its importance.

**PDR Feature and Image Feature Fusion.** We conduct three types of feature fusions, including input level, feature level, and output level fusion. The best performance is achieved by performing the feature level fusion of the PDR features and image features from our proposed encoders.

**Refinement.** The third group shows that the depth prediction performance can be improved by our RefineNet evenly either with or without GDC. Although the RefineNet can only slightly improve the performance of dense depth prediction, it can significantly improve the monocular 3D object detection task (+23%), demonstrating the effectiveness of the RefineNet. More qualitative and efficiency analysis of the RefineNet can be found in the supplementary materials.

## 5 Conclusion

We proposed a two-stage self-supervised framework FusionDepth that effectively fuses features from monocular camera images and sparse LiDAR data. Our method outperforms all the state-of-the-art methods on both depth prediction and completion tasks. Also shows a remarkable advantage for downstream tasks like monocular 3D object detection.

**Acknowledgments**

This work was supported in part by U.S. DOT UTC grant 69A3551747117.

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
