# OpenReview forum: "Advancing Self-supervised Monocular Depth Learning with Sparse LiDAR"
_robot-learning.org/CoRL/2021/Conference — CoRL2021 Poster_

### Official Review · Reviewer_SZdB · 2021-07-21

**Originality:** Fair
**Technical Quality:** Good
**Clarity Of Presentation:** Very Good
**Impact:** 3

**Recommendation:**

Weak Reject: I recommend rejecting the paper, but will not argue for my recommendation if the majority of other reviewers have a different opinion.

**Summary:**

This paper proposes a method for fusing sparse lidar point clouds with RGB images to obtain accurate dense depth predictions. Key to the performance of the approach is the use of pseudo-dense representations of the sparse lidar point cloud which assist the convolutional network in effectively fusing the data. The results show that the approach outperforms a range of current methods in terms of depth completion, monocular depth prediction and stereo depth prediction.


**Issues:**

I would like the authors to address the following points:
- How does the method perform in more challenging indoor scenes?
- I think the runtime results should be moved into the main paper as this is quite important for the reader to know


**Reviewer Expertise:**

Fair: Some knowledge of the area

**Strengths And Weaknesses:**

**Strengths**
- The paper is very well written and polished. The figures are nicely presented and help to explain the approach. The paper seems to be mature and is of publication quality should it be accepted.
- I think the approach is very interesting and well thought out. What I particularly like about it is that multiple sources, i.e. sparse depth, stereo supervision and monocular training are all combined elegantly into one framework. Many approaches can only train with monocular supervision which is quite limiting.
- The experimental evaluation is extensive and confirms the claims made in the paper. A good ablation study is done showing how each component contributes to the performance. The method is compared to a multitude of existing approaches. The proposed method performs very well compared to these. Including some other tasks such as mono-only and stereo prediction in the evaluation is useful as it allows the reader to place the method holistically (not just focussing on depth completion). This is definitely very useful as a comparison when making decisions for a real system. (However, see weakness below as I think this has been overdone a bit).
**Weakness**
- While the evaluation is quite extensive, I think the proposed approach is more of a depth completion approach for which there are a number of benchmarks (like the KITTI online depth completion benchmark) and many existing methods. However, the paper focuses extensively on monodepth methods in the comparison. This seems a bit misleading to me as it is obvious that mono approaches will perform worse. Maybe using this space to show the performance in other environments would be better. Also, why has the approach not been evaluated on the online benchmark?
- The approach has only been evaluated on KITTI where there is lots of repeated and regular structure. How does the method work in a general robotic setup such as would be the case for a home robot in an indoor environment? Or a robot in an outdoor field or sidewalk?
The improvement gained by the approach on the depth completion evaluation is quite small i.e. the existing methods already perform quite well.
- There is no comparison between the capacity of the proposed network and those in the literature. Does the method use significantly more layers or weights?


**Summary Of Recommendation:**

The paper itself is well written and I like both the manner in which the sparse lidar is effectively fused with the RGB and the natural manner in which multiple sources of supervision can be used. The main weakness though is that this only results in an incremental improvement on the (many) existing methods for depth completion. I do not see what sets this paper apart.

---

> ### Author Response · Authors · 2021-08-23
> **Response to Reviewer SZdB**
>
> Dear Reviewer:
>
> Thank you for your thoughtful and constructive feedback, we are glad to hear your acknowledgment of our model design and presentation, here we address the suggestions and concerns as follow:
>
> > 1.1. The proposed approach is more of a depth completion model?
>
> * Our method is very different from the depth completion task. The key difference is that depth completion methods normally take dense (i.e. 64-beams) LiDAR point cloud data as input, while the 4-beams LiDAR points are used as input to our model making the CNN feature extraction much more challenging. To address this sparsity issue, we proposed our PDR module to make the deep networks extract the features more effectively from the sparse LiDAR points.
> * Another major difference is that the depth completion model is normally **supervised** by dense point clouds aggregated from 11 frames of 64-beams LiDAR while our model is **unsupervised**. Our method is more similar to the depth prediction task [17, 19, 20, 21, 47, 52], which is usually unsupervised, has no access, or only a few other data modalities (LiDAR, Radar, Audio, etc.) Following the convention and the benchmark set by other methods [17, 19, 20, 21, 47, 52], our model is categorized as a depth prediction model.
>
> > 1.2. This seems a bit misleading to me as it is obvious that mono approaches will perform worse?
> * For comprehensive comparison with the state-of-the-art methods, we report the performance with different types of state-of-the-art models including monocular, stereo, supervised, unsupervised, and sparse-LiDAR-aided methods in Table 1. To further fairly compare with the state-of-the-art methods that use the same level of data/supervision as our model, we compared with the sparse-LiDAR-aided unsupervised depth prediction model [19, 20, 21] under different settings in Table 2. With the same amount of sparse-LiDAR as input to the dense depth prediction network, our model significantly outperforms the state-of-the-art sparse-LiDAR based model.
>
> > 1.3. Why has the depth prediction not been evaluated on the online benchmark?
> * All the existing methods that were evaluated on the KITTI depth prediction benchmark are vision-based which only takes monocular or stereo images as input. It is unfair to compare with these models since our model has access to extra data (sparse LiDAR). To fairly compare the state-of-the-art methods under the same setting, we evaluate our method on the validation set following other state-of-the-art LiDAR-Aided depth prediction methods [19, 20, 21, 47]. We added this information to Section 4.1 of the latest revision and highlighted it in red.
>
> * To demonstrate the quality of the predicted depths, we report the results for both low-level tasks (dense depth prediction) and high-level downstream tasks (monocular 3D object detection) and the result confirms that our model can learn better depth predictions.
>
> > 1.4. Why has the depth completion not been evaluated on the online benchmark?
>
> * Almost all the methods evaluated on the KITTI depth completion benchmark are trained in the supervised setting, with LiDAR points aggregated from multiple farmers. Since our models are trained in an unsupervised setting and only have access to very sparse LiDAR from one frame, therefore, it is unfair to directly compare these models on the leaderboard. By following the same setting proposed in the recent state-of-the-art work [29, 30, 31, 32, 33], we compared these unsupervised models under the same setting and achieved better performance. We have added this information to Section 4.3 of the revision and highlighted it in red.
>
> > 1.5. The detection online benchmark?
> * For the detection task, we evaluated on the KITTI detection online benchmark as reported in the paper because it is a fair comparison.
>
> > 2. How does the method work in a general robotic setup such as would be the case for a home robot in an indoor environment?
> * In this paper, we mainly verified the proposed method on the public benchmark for the autonomous driving scenario. Our model outperforms the state-of-the-art models on multiple benchmarks under the same setting. We believe our model can be transferred to other scenarios such as indoor datasets or other autonomous driving datasets. Due to the time and resource limitation, we will extend our work to more scenarios in the future.
>
> > 3. The capacity of the proposed network? And the runtime result should be moved to the main paper?
> * Our method uses the same backbone (ResNet-18) as the baseline model “Monodepth2 [17]”, we added this detail and the runtime results to the main paper Section 4.2 in revision and highlighted in blue.

---

### Official Review · Reviewer_vPeH · 2021-07-21

**Originality:** Good
**Technical Quality:** Very Good
**Clarity Of Presentation:** Good
**Impact:** 3

**Recommendation:**

Strong Accept: I recommend accepting the paper and will argue for my recommendation even if other reviewers hold a different opinion.

**Summary:**

This paper predicts depth images based on monocular RGB images and sparse LIDAR data, by fusing both in there so called DepthNet, which is afterwards improved by RefineNet architecture, which uses the result, the features from the sparse LIDAR data and the sparse LIDAR data itself. They show in a extensive evaluation section superior results to prior state-of-the-art methods in different categories. Furthermore, they show how each single change affects the final result in an depth ablation study.

**Issues:**

* Results for Pseudo LIDAR++ with stereo have to be included
* The depth completion and prediction methods should be evaluated on the test set not the validation set and furthermore the values should be in the same format as in the online leader board, to make it easier for everyone to understand if the claim about state of the art performance is correct
* The related work section should contain more explanation on how this method compares to existing methods and not just list them.

**Reviewer Expertise:**

Very good: Comprehensive knowledge of the area

**Strengths And Weaknesses:**

Strengths:
* the paper is well written and explains all important concepts in depth
* present a novel fusion mechanism by reusing the features from the first network in the second to refine the depth image
* they claim to be the new state of the art method for depth prediction
* Extra point for including a point cloud result, as no one can properly judge how well a depth completion method works by looking at a colored image

Weaknesses:
* The related work section, while citing many papers lacks a deeper comparison to these prior methods, to better clarify the difference between those works and this new work
* Pseudo Dense Representation: Here they explain how they generate a dense depth map based soley on the sparse LIDAR data. However, they don't answer what happens if more than one sparse data point is close to an evaluated position, this should be added. Is the last point used or is interpolated between the two?
* Table 1 contains a lot of comparisons to methods, which use different input data. 22 rows use only monocular or stereo data and don't use any sparse LIDAR data as an input, which makes the comparison a bit unfair.
* I am furthermore confused by the claim that they beat state of the art, if I check out the leaderboard and try to compare it. I already have the problem that they don't provide the results on the test set but on the validation set, for example they report for "Sparse to dense [30]" an iRMSE of 4.28, while in the leaderboard: http://www.cvlibs.net/datasets/kitti/eval_depth.php?benchmark=depth_completion it says 2.80. I don't understand why they didn't evaluate the test data online, to make it easier to compare it existing methods.
* Furthermore, do I struggle with the results report for RSME, as they seem much closer to iRSME values than to actual RSME values for KITTI.
* Lastly, why are the results for the stereo case of Pseudo Lidar++ not reported, I agree that it makes it more fair to only use monocular to make the comparison more fair. That does not mean though that one should not report the original values, because now I can't judge if this change complete breaks the performance of this algorithm and then gives you edge or if you would have beaten it even with stereo? For me to accept this paper I need to know these values to access if changing Pseudo Lidar++ to monocular is something which the architecture can handle or if you broke Pseudo Lidar++ on purpose to give your method an advantage. As we can only judge other methods as they published them not as we want them to be.

**Summary Of Recommendation:**

I am mostly basing my review on my doubts about the evaluation. If the authors can prove or explain the choices they made in their evaluation then I can gladly switch my recommendation to strong accept. But, at the moment I feel like that the evaluation was designed in a way to make it harder to compare it to the leaderboard and weaken on purpose the other approaches to boost their own method.

Based on that I can not recommend accepting this paper in its current form to CORL.

After rebuttal:
The authors answered most of my problems with the evaluation and henceforth I have no objection anymore against accepting this paper to CORL. They adjusted the evaluation and positioned themselves better against Pseudo LiDAR++. They also explained why an evaluation on the validation set is justified and extended the related work.

---

> ### Author Response · Authors · 2021-08-23
> **Response to Reviewer vPeH (Part 1)**
>
> Dear Reviewer:
>
> Thank you for your thorough and helpful comments. We greatly appreciate your acknowledgment of our technical quality, novelty, and presentation. You brought up some great questions and concerns, please find our answers below:
>
> > 1. The related work section lacks a deeper comparison.
> * Due to the space limitation, we only briefly introduced and compared the related approaches in the “related works” section. We have added more discussion about the internal connection among these tasks in the revision and highlighted it in blue in the Related Work Section.
>
> > 2. What happens if more than one sparse data point is close to an evaluated position?
> * If more than one sparse data point is close to an evaluated position, the confidence and depth scores generated from these multiple data points will be averaged for the PDR. We added this detail to Section 3.1 of the revision and highlighted it in blue.
>
> > 3. Many of the methods in Table 1 are vision-only methods?
> * The majority of the existing methods for depth prediction are vision-only based which mainly takes monocular or stereo images as input. As a rising trend, more and more methods were recently proposed to utilize the sparse-LiDAR as an extras signal to guide the model to produce better depth. However, as a newly rising research direction, there are only a handful of methods [1, 19, 20, 21, 47] that use sparse LiDAR data for depth prediction and all of them are included in our paper. To make the comparison more comprehensive and show the significance of utilizing sparse LiDAR, we also have some vision-only methods for reference in Table 1, we have reduced them from 22 lines to 8 lines.
> * To fairly compare with these state-of-the-art sparse-LiDAR based methods under different settings (such as with different numbers of sparse-LiDAR points and different ways to sample sparse-LiDAR points), we reported the results and compared to them [1, 19, 20, 21, 47] in Table 2 and our model significantly outperforms all other methods under these different settings. These results confirm the capacity and potential of our proposed method.
>
> > 4.1. Inconsistent results for “Sparse to Dense [30]”?
> * That iRMSE value of 2.80 from KITTI Leaderboard is under **supervised** setting (using accumulated 11 frames of the 64-beams LiDAR points as ground truth supervision) while our method is **unsupervised** and can not be directly compared. Under the same unsupervised setting, “Sparse to Dense [30]” reported 4.32 in Table 3, and “Self-Deco [29]” reported 4.28 in Table 3”. Our model achieves 3.382 which is much better than those two methods under the same setting.
>
> > 4.2. Why do not evaluate the online depth completion benchmark?
> * Most (over 95%) of the methods on KITTI depth completion Leaderboard is under supervised training with aggregated multiple frames of LiDAR point clouds as input, however, our method is unsupervised and only takes very sparse LiDAR points as input. It is unfair to compare our model with these methods on the leaderboard. Following the convention and the setting of other state-of-the-art self-supervised depth completion methods [29, 33], we evaluated our completion model on the validation dataset for a fair comparison.  We added this information to Section 4.3 of the revision and highlighted it red.
>
>
> > 5. RMSE seems much closer to iRSME values than to actual RMSE values for KITTI?
> * The RMSE and iRMSE on the KITTI depth completion leaderboard use the unit of millimeter and kilometer, while metrics in the depth prediction is using meters. All evaluations in our paper are following recent state-of-the-art papers [1, 19, 20, 21, 29, 30, 31, 33 47].

---

> > ### Author Response · Authors · 2021-08-23
> > **Response to Reviewer vPeH (Part 2)**
> >
> > > 6. Results for Pseudo LiDAR++ not reported?
> > * The detection mAP (car, $AP_{3d}@0.7$, moderate) result of “Pseudo LiDAR++ [1]” is 63.8 from their paper, we will add it to the revision paper to demonstrate the upper bound performance, but it can not be directly compared to our method, reasons are as follows:
> >
> > * The “Pseudo LiDAR++ [1]” method consists of two parts, the depth module, and the post-processing GDC module. The Depth module is adapted from the PSMNet (Chang & Chen, 2018) which takes stereo images as input and predicts initial dense depth maps. While the GDC module uses the sparse LiDAR data to correct the depth prediction from the Depth module, to improve the downstream detection accuracy.
> >
> > * Their Depth model is stereo, supervised (which can greatly improve the detection accuracy), pre-trained on scene flow dataset, and then fine-tuned on KITTI detection dataset. While our model is monocular, self-supervised, trained only on the KITTI Depth dataset, has no access to the detection set. Among those differences, the key differences are that PL++ is a **supervised** method having access to **stereo frames**, while our model is **unsupervised** with a **monocular** image as input. Normally the stereo-based methods perform much better than the monocular-based method. Therefore, with these differences in the settings, it is not fair to directly compare the performance of our model with the original PL++ model.
> >
> > * For a fair comparison, in our experiment, we replaced the supervised stereo depth module with the state-of-the-art unsupervised monocular depth network “Monodepth2 [17]” to make the comparison fair, and left the key GDC module unchanged. In this way, our model and the PL++ would have the same level of input data and supervision making the comparison fair. The key contribution of the PL++ is the GDC module while we did not change this part, therefore, we did not take any advantage of the “Pseudo LiDAR++ [1]”. We also revised the paper to call this model Mono PL++ for clarification.
> >
> > * We also revised the paper in Section 4 to call this model Mono PL++ for clarification, highlighted in red and blue.
> >
> > > 7. Why is depth prediction not evaluated on the online leaderboard?
> > * KITTI depth prediction Leaderboard only contains camera-based methods, it is unfair to evaluate our method on that leaderboard because we have access to sparse LiDAR data. Following the convention and settings in other sparse-LiDAR based methods [19, 20, 21, 47],  we evaluate our method on the validation set. Our model significantly outperforms these methods under the same settings indicating the potential of our proposed model. We added this information to Section 4.1 of the latest revision and highlighted it in red.

---

> > ### Comment · Reviewer_vPeH · 2021-08-31
> > **Response**
> >
> > Thanks for your response, adding those details to the related work makes it worth reading.
> >
> > The evaluation now is more focused on the contribution and makes it clear how to position it against only vision based methods.
> >
> > Thanks again for providing a detailed explanation, I am not super happy with the presentation of Pseudo LiDAR++, but I don't know how to do it better. I see the struggle that there are not enough methods around there to compare, so I appreciate the effort of increasing the range of compared methods.
> >
> > One las thing, I would highlight the difference in more depth under Table 3 and also include the fact that PL++ is supervised while your approach is not. Just to make it more apparent to the reader, that this comparison is not fair.

---

### Official Review · Reviewer_y8K7 · 2021-07-23

**Originality:** Good
**Technical Quality:** Good
**Clarity Of Presentation:** Very Good
**Impact:** 3

**Recommendation:**

Weak Accept: I recommend accepting the paper, but will not argue for my recommendation if the majority of other reviewers have a different opinion.

**Summary:**

This paper proposes an architecture and self-supervised learning approach for dense monocular depth estimation when a very sparse set of accurate depth readings are available from a LiDAR. The proposed architecture is a multi-stag architecture that consists of a network that receives input images and densified sparse LiDAR readings and produces and inital depth map. The initial depth maps is refined using a refinement network that essentially distills the GDC depth correction from PseudoLIDAR++ into an efficient network. Training is done using self-supervision based on photometric cues and the output of the offline depth correction module. Experiments are shown on KITTI, including results on a 3D object detection downstream tasks.

**Issues:**

Minor issues:

- l. 92: Previous text refers to [1] as Pseudo-LiDAR++, but here suddenly as PL++ without an introduction of the abbreviation.

- Table 1: "excepctionall time-consume" - broken sentence.

- Table 2 is hard to read because of small fonts in combination with misaligned table rulers.

**Reviewer Expertise:**

Good: General knowledge of the area

**Strengths And Weaknesses:**

Strengths:

- Seemingly good results at very high frame-rates. However, I do have some questions about the results (see below).
- I particularly like that experiments on a down-stream tasks are shown. The improvement shown there is impressive.
- Reasonable architecture construction that consolidates the individual stages that have been layed out in Pseudo-LiDAR++ into a single efficient end-to-end pipeline.


Weaknesses:

- The comparisons in Table 1 are not quite clear to me. For example the BTS is shown with \delta_1 of 0.904, whereas the original paper shows results somewhere in the range of 0.95 (depending on the employed backbone) - this would actually outperform the proposed work without any LiDAR points required. Similarly DORN in the original paper also shows values of up to 0.93, but the table here only lists it with 0.897. I'd be grateful if the authors could clarify the discrepancy here.

- This is unfortunately shared by many works in this space: the approach is only evaluated on KITTI. No other datasets are shown. Self-supervised approaches promise, in principle, that they can be deployed anywhere (in this work of course conditioned on the availability of LiDAR measurements). It would be great to go beyond KITTI and actually show off the flexibility of this approach. This could be done by showing results on transfer or fine-tuning on other datasets (for example Apolloscape) rather than squeezing every last drop of performance out of KITTI.

Other:

- Is the pose network really required? It seems that temporally consecutive LiDAR point clouds could be aligned using standard geometric approaches to get the pose. This procedure would be guaranteed to be generalization, unlike a pose network which are known to be hard to generalize (have you tried playing a sequence backward? - many pose networks fail to generalize even to this simple task)



**Summary Of Recommendation:**

This work provides a technically well executed pipeline that seems to show good results, especially on the downstream task, which I believe deserve to be seen. However, I do have questions about the evaluation which I hope the authors can resolve.

-- Post rebuttal --
The authors resolved my major questions. What remains is the somewhat evaluation only on KITTI. I do believe however that this shouldn't be a reason for rejection. I thus keep my initial rating.

---

> ### Author Response · Authors · 2021-08-23
> **Response to Reviewer y8K7**
>
> Dear Reviewer:
>
> Thank you for acknowledging the strong performance of this work and the quality of the presentation. You brought up some great questions and suggestions, here we address them as follows:
>
> > 1. Clarification about the results of other models including BTS and DORN.
> * The KITTI depth prediction dataset Eigen split has two versions of the testing data: **Original Testing Data** which use raw LiDAR points as ground truth and **Improved Testing Data** which removes some points. The depth prediction models normally achieve much better performance on the Improved Testing Dataset as we stated in the caption of Table 1. Following the state-of-the-art models, we mainly report the results on the Original Testing Dataset since it is more widely used.
>
> * For the BTS and DORN model, their delta_1 result on the Original Test Data is 0.904 and 0.897, while the result on the Improved Test Data is 0.95 and 0.93. You can find the detailed results from the following resources:
>
> Table 2 in the paper:
> ```
> Shu, Chang, et al. "Feature-metric loss for self-supervised learning of depth and egomotion." European Conference on Computer Vision. Springer, Cham, 2020.
> ```
> Table 4 in the paper:
> ```
> Gonzalez, Juan Luis, and Munchurl Kim. "PLADE-Net: Towards Pixel-Level Accuracy for Self-Supervised Single-View Depth Estimation With Neural Positional Encoding and Distilled Matting Loss." Proceedings of the IEEE/CVF Conference on Computer Vision and Pattern Recognition. 2021.
> ```
> Table 3 in the paper:
>
> ```
> Guizilini, Vitor, et al. "3d packing for self-supervised monocular depth estimation." Proceedings of the IEEE/CVF Conference on Computer Vision and Pattern Recognition. 2020.
> ```
>
> > 2. Evaluate on other datasets?
> * Our proposed model is a general depth prediction model and all the modules in our model can be applied to datasets from other settings such as indoor/outdoor datasets. Due to the limit of time and computation resources, we only performed experiments on the autonomous driving setting, and we thoroughly evaluated our method on 3 different tasks and demonstrated the capability of our proposed model. We will continue to transfer our model to more datasets in the future, including the Apolloscape as you suggested.
>
> > 3. Use geometric approaches to get the pose from consecutive LiDAR point clouds?
> * This is a very good point, we can infer relative poses from the consecutive LiDAR point cloud frames with geometric methods like the ICP, and this will be more interpretable. But due to the extreme sparsity of the 4-beams LiDAR, we are not sure if this method can produce poses with high precision. Actually, we have tried to transform one source frame of the LiDAR point cloud to a neighboring target frame with the pose predicted from the PoseNet, and it aligned pretty well with the target frame point cloud. In the future, we will explore the possibility of combining the PoseNet and these geometric methods for better pose prediction.
>
> > 4. Other minor issues of writing.
> * We have addressed the remaining minor suggestions in the revision. Appreciate your detailed suggestions.

---

### Official Review · Reviewer_HJoT · 2021-07-23

**Originality:** Good
**Technical Quality:** Good
**Clarity Of Presentation:** Good
**Impact:** 3

**Recommendation:**

Weak Accept: I recommend accepting the paper, but will not argue for my recommendation if the majority of other reviewers have a different opinion.

**Summary:**

This paper proposes a pseudo-Lidar framework from sparse Lidar points and an RGB camera. A two-stage network is utilized which preprocesses the sparse Lidar points and fuses them with RGB information to obtain a coarse depth prediction and then refines this. During training, multiple time frames are used to obtain additional losses to compensate the lack of supervised depth data.

**Issues:**

- more discussion on the reason of network module selection.
- explanation of the difference between the dataset benchmark evaluation and your reported results.

**Reviewer Expertise:**

Good: General knowledge of the area

**Strengths And Weaknesses:**

Although, categorically monocular 3D object detection includes pseudo-lidar based methods, once the depth information is included (sparse or dense), it is not monocular camera estimation anymore, it includes at least two sensors for prediction. This is a repeated mistake in many papers.


If multiple frames can be used, then can't their sparse depth correspondences aggregated over time to produce a denser point cloud which could help depth correction as well?

For depth prediction, did you submit your results to original KITTI page?

What is the difference between your reported results for PseudoLidar++ and KITTI page, can you explain briefly in the paper? (For all tables.)

Minor points:

Our experiments demonstrates-> Our experiments demonstrate

**Summary Of Recommendation:**

The authors have compared a number of methods. Their own proposal, which is a combination of multiple networks, is reported to achieve a high accuracy. Although I could not check it in the original submission page of the datasets. The codes are also provided, which I appreciate. However, I did not have time to check the codes.

---

> ### Author Response · Authors · 2021-08-23
> **Response to Reviewer HjoT**
>
> Dear Reviewer:
>
> Thank you for your positive and constructive feedback, we are pleased to hear that you found our advance in originality and convincing technical quality. Here we answer your great questions and suggestions as follows:
>
> > 1. Once the LiDAR information is included (sparse or dense), it is not monocular camera estimation anymore?
> * Our model belongs to the sparse-LiDAR aided methods category since our model takes both monocular image and sparse-LiDAR as inputs. To fairly compare with the state-of-the-art methods, we reported the performance and compared with the state-of-the-art LiDAR-Aided methods [1, 19, 20, 21, 47] on the KITTI dataset. With the same setting, our model significantly outperforms the state-of-the-art methods. Relying on a low-cost sparse LiDAR device, our model produces much better depth and has the potential to be deployed to various autonomous robot/vehicle scenarios.
>
> > 2. Aggregate multiple frames of point clouds?
> * This is a very good point. We have tried to fuse two frames of the sparse LiDAR points with the pose prediction from PoseNet, but it did not provide significant improvements. One challenge is that PoseNet does not provide the actual scale of the camera translation, and another challenge is that the object motion across frames coupled into these LiDAR points may bring noises to the network. Using temporal multi-frame data is a promising direction, actually, it is our focus on the future work.
>
> > 3. For depth prediction, did you submit your results to the original KITTI page?
> * For depth prediction, we did not submit to the KITTI benchmark because all the methods reported in the leaderboard are vision-only methods (only taking monocular or stereo images), it is unfair to directly compare our method with them. Following the setting of the sparse-LiDAR aided methods [1, 19, 20, 21, 47], we report the results on the KITTI validation dataset and compare with them to demonstrate our method's capability. For the monocular 3D object detection task, we tested our model on the online testing server as other models and reported the results in the paper.
>
> > 4. What is the difference between your reported results for PseudoLidar++ and the KITTI page?
> * The original “Pseudo LiDAR++ [1]” can not be fairly compared to our method, we made some changes on the Pseudo LiDAR++ model [1] for fair comparison:
>
> * The “Pseudo LiDAR++ [1]” method consists of two parts, the Depth module and the GDC module. The Depth module is adapted from the PSMNet (Chang & Chen, 2018), taking stereo frames as input. While the GDC module is a post-processing module that uses the sparse LiDAR data to correct the depth prediction from the Depth module, to produce higher-quality dense depth.
>
> * The key difference between our setting with the original PL++ is that PL++ is a supervised method having access to stereo frames, while our model is unsupervised with a monocular image as input. Normally the stereo-based methods perform much better than the monocular-based method. Therefore, with these differences in the settings, it is not fair to directly compare the performance of our model with the original PL++ model.
>
> * To fairly compare with the PL++ method under the same setting, in our experiment, we replace the stereo depth model with the same monocular depth model (Monodepth2 [17]) as our model and reported the performance on both dense depth prediction task and monocular 3D object detection task. With this change, the two models have access to the same level of data and supervision, therefore, the performance can be fairly compared. To make the paper more clear, we changed the PL++ to monocular PL++ (MonoPL++) in all the Tables and the main paper.
>
> > 5. More discussion on the reason for network module selection.
> * Due to the page limitation, most of the contents regarding the module selections are included in Section 3 of the supplementary materials. We will move part of them into the main paper to make the paper more clear.

---

> > ### Comment · Reviewer_HJoT · 2021-08-30
> > **Re-review**
> >
> > The points related to benchmark and the modifications in other methods should be mentioned with its reasons in the paper.

---

> > > ### Author Response · Authors · 2021-08-31
> > > **Reply to Reviewer HjoT**
> > >
> > > Dear Reviewer:
> > >
> > > Thank you for your reply, we added this information to Section 4.1 and 4.3 and highlighted it in red in the latest revision.

---

### Meta-Review · Area_Chair_VNs8 · 2021-08-13

**Recommendation:** Accept (Poster)
**Confidence:** 5

**Metareview:**

The paper proposes an approach for self-supervised depth estimation from sparse LiDAR scans. The reviewers find the paper to be generally well written and well structured. The reviewers appreciate the experiments on downstream tasks and the good results. Most of the major concerns raised by the reviewers have been addressed in the rebuttal, except for the benchmarking on the KITTI test set. For this the authors have provided the reasoning on why its unfair to compare to methods on the leaderboard. I agree to an extent but it would still be nice to see how well the method performs, getting a lower metric value on the leaderboard will not diminish the contributions of the paper. Nevertheless, I recommend acceptance and I encourage the authors to still benchmark their approach on the KITTI depth completion or depth prediction test set.

---

> ### Author Response · Authors · 2021-08-24
> **Response to Meta Reviewer VNs8**
>
> Dear Meta Reviewer:
>
> Thank you for your thorough review and constructive feedback. We greatly appreciate your acknowledgment of our written, structure and the result of extensive experiments. We have addressed all reviewer’s concerns and responded separately, here we briefly summarize our response for the major concerns as follows:
>
> > 1. Lack of results on the KITTI test set:
> * For 3D monocular detection, we followed the state-of-the-art methods and evaluated our method on the KITTI online server, and compared it with them in our paper.
> * For depth prediction, All the existing methods that were evaluated on the KITTI testing benchmark are vision-based which only takes monocular or stereo images as input. It is unfair to compare with these models since our model has access to extra data (sparse LiDAR). To fairly compare the state-of-the-art methods under the same setting, we evaluate our method on the validation set following other state-of-the-art LiDAR-Aided depth prediction methods [19, 20, 21, 47].
> * For depth completion, almost all the methods evaluated on the KITTI depth completion benchmark are trained in the supervised setting, with LiDAR points aggregated from multiple farmers. Since our models are trained in an unsupervised setting and only have access to very sparse LiDAR from one frame, therefore, it is unfair to directly compare these models on the leaderboard. By following the same setting proposed in the recent state-of-the-art works [29, 30, 31, 32, 33], we compared our model with these unsupervised models under the same setting and achieved better performance.
> * We have added this information to the section 4.1 and 4.3 and highlighted it in red in the latest revision.
>
>
> > 2. Clarity on the reported metrics:
> * We clarified our evaluation metrics to the reviewers, it is mainly about the difference in units. All metrics used in our paper follow the convention set by other state-of-the-art papers.
>
> > 3. The discrepancy with the values of compared methods:
> * These discrepancies come from the different testing datasets (KITTI original testing data vs KITTI improved testing data) and different training settings (supervised vs self-supervised). We have clarified all these concerns to the reviewers, and added these details in the latest revision, highlighted them in blue and red.
>
> > 4. Comparisons with more relevant methods than simple monocular approaches:
> * The majority of the existing methods for depth prediction are vision-only which mainly takes monocular or stereo images as input. As a rising trend, more and more methods were recently proposed to utilize the sparse-LiDAR as an extras signal to guide the model to produce better depth. However, as a newly rising research direction, there are only a handful of methods [1, 19, 20, 21, 47] that use sparse LiDAR data for depth prediction and all of them are included in Table 1. To make the comparison more clear, we removed many of the vision-only methods from Table 1 in the revision.
> * To fairly compare with these state-of-the-art sparse-LiDAR based methods under different settings (such as with different numbers of sparse-LiDAR points and different ways to sample sparse-LiDAR points), we reported the results and compared with them [1, 19, 20, 21, 47] in Table 2 and our model significantly outperforms all other methods under all these settings. These results confirm the capability and potential of our proposed method.
>
> > 5. More insights on the network design and complexity:
>
> * Due to the page limitation, most of the contents regarding the module selections are included in Section 3 of the supplementary materials.
>
> * Our method uses the same backbone (ResNet-18) as the baseline model “Monodepth2 [17]”, we have added this detail and the runtime results to the main paper Section 4.2 in revision and highlighted it in blue.

---

### Decision · Program_Chairs · 2021-09-13

**Decision:**

Accept (Poster)

**Comment:**

The paper proposes an approach for self-supervised depth estimation from sparse LiDAR scans. The reviewers find the paper to be generally well written and well structured. The reviewers appreciate the experiments on downstream tasks and the good results. Most of the major concerns raised by the reviewers have been addressed in the rebuttal, except for the benchmarking on the KITTI test set. For this the authors have provided the reasoning on why its unfair to compare to methods on the leaderboard. I agree to an extent but it would still be nice to see how well the method performs, getting a lower metric value on the leaderboard will not diminish the contributions of the paper. Nevertheless, I recommend acceptance and I encourage the authors to still benchmark their approach on the KITTI depth completion or depth prediction test set.